# Opioids and Cancer: Current Understanding and Clinical Considerations

**Dhananjay Sah** [1,2], **Hagit Shoffel-Havakuk** [3,4], **Nir Tsur** [3,4], **Megan L. Uhelski** [5], **Vijaya Gottumukkala** [1] and **Juan P. Cata** [1,2,*]

1    Department of Anesthesiology and Perioperative Medicine, The University of Texas MD Anderson Cancer Center, Houston, TX 77030, USA; dksah@mdanderson.org (D.S.); vgottumukkala@mdanderson.org (V.G.)
2    Anesthesiology and Surgical Oncology Research Group (ASORG), Houston, TX 77030, USA
3    Department of Otolaryngology-Head and Neck Surgery, Rabin Medical Center, Petach Tiqva 4941492, Israel; hagitsho@clalit.org.il (H.S.-H.); nirt@clalit.org.il (N.T.)
4    Faculty of Medicine, Tel Aviv University, Tel Aviv 6997801, Israel
5    Department of Pain Medicine, The University of Texas MD Anderson Cancer Center, Houston, TX 77030, USA; mluhelski@mdanderson.org
*    Correspondence: jcata@mdanderson.org

**Abstract:** Pain is one of the most common symptoms in patients with cancer. Pain not only negatively affects the quality of life of patients with cancer, but it has also been associated with reduced survival. Pain management is therefore a critical component of cancer care. Prescription opioids remain the first-line approach for the management of moderate-to-severe pain associated with cancer. However, there has been increasing interest in understanding whether these analgesics could impact cancer progression. Furthermore, epidemiological data link a possible association between prescription opioid usage and cancer development. Until more robust evidence is available, patients with cancer with moderate-to-severe pain may receive opioids to decrease suffering. However, future studies should be conducted to evaluate the role of opioids and opioid receptors in specific cancers.

**Keywords:** opioids; cancer biology; opioids and cancer biology; opioids and immune function; opioids; cancer

## 1. Introduction

The International Association for the Study of Pain defines pain as "an unpleasant sensory and emotional experience associated with, or resembling that associated with actual or potential tissue damage [1]." Pain affects over a third of patients with cancer after curative treatment, more than half of them during anticancer therapies, and over two-thirds of those with advanced malignancies [2]. Pain also affects the survival of patients with active cancers [3,4]. It has been shown that pain intensity is an independent predictor of 5-year survival in patients such as those with head and neck cancers and advanced non-small-cell lung cancers [3]. Based on those premises, pain in patients with cancer should be treated with a multimodal approach in which opioids remain an essential component.

In 1803, morphine was extracted from opium [5]. Paul Janssen synthesized fentanyl in 1960 [6].

Opioids have been indicated for the treatment of cancer pain since the 1950s [7]. Prescription opioids, including drugs such as morphine, oxycodone, fentanyl, buprenorphine, and methadone, are indispensable in the management of severe pain, especially in oncological patients. Currently, the American Society of Clinical Oncology recognizes that "opioids are the first-line approach for moderate to severe chronic pain associated with active cancer." The society also recommends that opioids should be prescribed to selected cancer survivors with post-cancer or treatment pain syndromes [8,9].

Opioids vary in their origin, either naturally extracted, modified, or semi synthetics, which affects their potency and adverse event profile, with morphine serving as the benchmark for opioid effectiveness due to its well-documented pharmacokinetic profile [10]. Oxycodone is noted for its oral bioavailability and prolonged pain relief, while fentanyl is distinguished by its extreme potency, making it effective for severe pain management in cancer patients [11,12]. The distinct chemical structures of these opioids determine their interaction with µ-opioid receptors (MORs) in the central nervous system, influencing both pain perception and emotional responses. Each opioid's unique properties, such as buprenorphine's partial agonistic effect and methadone's NMDA receptor antagonism, necessitate careful prescription and monitoring to balance efficacy against potential risks like respiratory depression and addiction [11,12]. Although opioids are commonly used in pain management, they are not without harmful consequences. Both acute and chronic treatment with opioids can result in side effects, including hyperalgesia, analgesic tolerance, withdrawal syndrome, neuroinflammation, and respiratory depression [13].

The potential impact of opioids and MOR on cancer outcomes has garnered attention recently. Research shows that MOR is in some cancer cells and cells regulating the tumor microenvironment [14–19]. Opioids can affect tumor growth and metastasis [18,20,21]. The exact pathways and effects of MOR signaling on cancer outcomes are currently being studied [4,14,22,23].

This manuscript aims to summarize our current understanding of the role of MOR and opioids in cancer progression.

## 2. MOR Biology in Cancer

Opioids act on the classical opioid receptors: µ, κ, and δ receptors. However, opioids can also bind to other receptors (i.e., the opioid growth factor receptor or bradykinin receptors). MOR is a G protein-coupled receptor (GPCR) located in the cytoplasmic membrane or nucleus of cells [24–27]. Activation of MOR includes (1) binding of the agonist to MOR-1, (2) decoupling of the receptor from the heterotrimeric inhibitory G protein (Gi), and (3) Gi-promoting an influx of $Na^+$ and $Ca^{2+}$ and the influx of $K^+$ via cAMP/PKA [28]. $\beta$-arrestin recruitment after MOR activation appears to contribute to some of the unwanted effects of classical opioids [29,30]. Beta-arrestins can trigger signaling cascades independently of G proteins, a phenomenon called "biased signaling".

When MOR is activated, it also signals two important intracellular pathways: the PI3K/AKT pathway and the Mitogen-Activated Protein Kinase (MAPK) route (Figure 1) [31,32]. With the activation of the PI3K/AKT pathway, the kinase AKT regulates several cellular activities, including growth, survival, and metabolism [33]. These kinases are also involved in the phosphorylation of transcription factors that affects cell survival, proliferation, and differentiation [33,34]. In some but not all cancer cells, MOR activation increases invasiveness and metastatic capabilities, suggesting its role in disease progression [18,35,36].

The binding of opioids with MOR activates multiple signaling pathways, leading to the promotion of cancer through tumorigenesis, angiogenesis, migration, metastasis, and EMT. Activated receptors affect CAMKII, STAT3, PKA, and RAS-HIF-1$\alpha$-GAB1-PI3K-AKT-cMyc, and phosphorylation of Src/HIF-1$\alpha$-Grb1-Grb2-PI3-Akt-GSK-3$\beta$ promotes tumorigenesis. $Ca^{2+}$-NO-MAPK/ERK, cAMP-ICAM, and PI3K-AKT-cMyc result in tumor angiogenesis and tumorigenesis. Acting on the EGFR-Src-GAB1/2-PI3K/Akt pathway promotes metastasis.

MOR expression in malignancies such as lung cancer is well documented and affects prognosis [37]. In human non-small-cell lung cancer, MOR overexpression promotes Akt and mTOR activation, tumor growth, and metastasis [38]. This is achieved by activating intracellular signaling pathways such as MAPK/ERK [39]. MOR expression in prostate cancer cells may affect tumor development [40]. According to a study, MOR activation promotes tumor growth and apoptosis resistance, which could affect treatment. Colorectal cancer cells express MORs, suggesting a role in disease progression [19,41]. Even aggressive pancreatic cancer expresses the MOR gene [42,43]. MOR in pancreatic cancer cells

modulates neurotransmitter production, which may affect tumor–environment nervous and immune system interactions. The presence of MORs in ovarian cancer cells stimulates their growth and may induce resistance to chemotherapy [35,44,45].

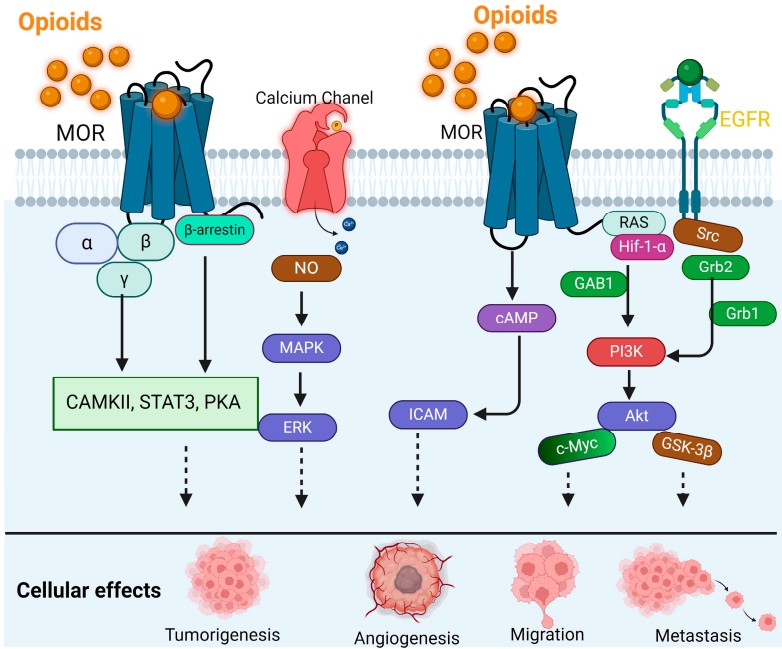

**Figure 1.** MOR signaling in cancer cells.

## 2.1. MOR and Cancer–Nerve Interaction

Cancer–nerve interactions have been highlighted as an essential factor in tumor development and spread [46,47]. The tumor microenvironment can promote neuroinflammation, and the addition of opioids that activate MOR expressed within the tumor and surrounding tissue can potentially stimulate tumor growth and further increase inflammation [48]. Perineural invasion (PNI), wherein cancer cells approach and invade the peripheral nervous system as tumors develop, is a marker for a poorer prognosis in multiple cancer types, including head and neck squamous cell carcinoma [49–51], prostate cancer [52], and pancreatic cancer [53]. This process can involve signals from cancer cells that stimulate nerve outgrowth, including neurotrophic growth factors, as well as signaling in the nerve that promotes cancer invasion [47]. In vitro, pancreatic adenocarcinoma cells were shown to induce neurotropism in cultured mouse dorsal root ganglion neurons, and enhanced neurite outgrowth was also associated with greater cancer cell colony growth [53]. PNI is associated with higher pain intensity [54], and pain intensity is also a predictor of poor prognosis in head and neck cancers [3] and non-small cell lung cancers [4].

In some cancers, MOR expression and activity are linked to PNI processes (Figure 2). Higher MOR expression was associated with a higher rate of PNI in pancreatic adenocarcinoma, and patients with higher MOR expression and high opioid consumption had lower overall survival, suggesting that interplay between pancreatic cancer cells and peripheral nerves in the context of MOR activity in the tumor microenvironment fosters tumor growth [42]. In ovarian cancer patients, higher MOR expression in the tumor was also associated with higher rates of PNI and higher pain intensity on post-operative day 1 following debulking surgery, but it was not predictive of overall survival [45].

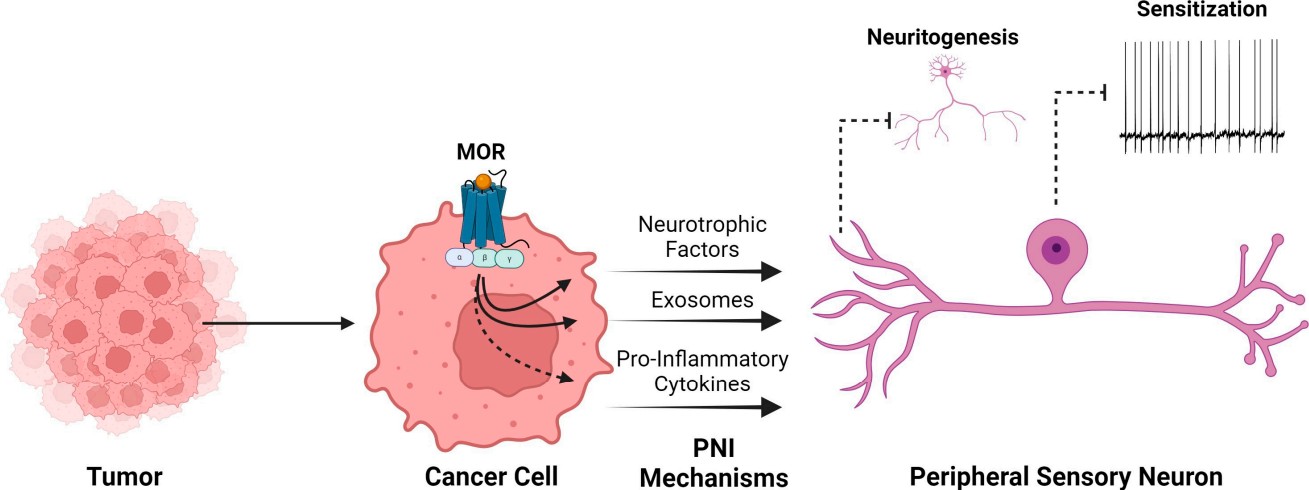

**Figure 2.** MOR expression in cancer cells contributes to mechanisms of perineural invasion.

A variety of factors released by cancer cells contribute to a microenvironment that promotes perineural invasion and peripheral sensory neuron sensitization. These include neurotrophic factors that influence neurite outgrowth and axonal guidance, exosomes whose contents include miRNAs that can promote hyperexcitability, and pro-inflammatory cytokines such as IL-6 that can directly sensitize neurons. The expression of MOR in cancer cells is linked to these impacts. When MOR was silenced in head and neck cancer cells (HNSCCs), neurite outgrowth was inhibited, and co-cultured trigeminal ganglion neurons did not develop neuronal sensitization that was found when MOR-expressing HNSCCs were present. Thus, downstream signaling from MOR influences the release of neurotrophic factors and exosome content and may alter the production of pro-inflammatory cytokines.

In an orthotopic xenograft model of head and neck cancer using Fadu HNSCCs, trigeminal ganglion neurons cultured from mice bearing tumors expressing MOR exhibited more robust sensitization (ectopic activity, low current thresholds, and more depolarized membrane potential) compared to neurons from mice that had tumors consisting of MOR knockout Fadu cells or those without tumors [14]. Co-culture of naïve mouse trigeminal neurons with the MOR knockout Fadu cell line inhibited neurite outgrowth compared to un-treated neurons or those co-cultured with MOR-expressing cancer cells, suggesting that the absence of MOR activity in Fadu cancer cells alters the tumor microenvironment to reduce neurotropism, potentially impacting the development of perineural invasion in vivo [14]. Overexpression of MOR in non-small-cell lung cancer cells was associated with enhanced tumor progression and higher Akt and mTOR activation [38]. Inhibition of Akt, mTOR, or MOR in this cell line suppressed cell proliferation, migration, and invasion. Among mTOR downstream effectors, the MNK-eIF4E signaling pathway and increased eIF4E-mediated translation have also been targeted for their role in pain modulation and chronic pain development, including chemotherapy-induced neuropathic pain [55]. Neuropathy associated with chemotherapy treatment is notoriously difficult to treat effectively and remains a primary reason for patients to discontinue treatment [56]. Elevated eIF4E has also been identified in uterine biopsies found to contain high-grade dysplasia or carcinoma [57].

MOR expression and activity changes have also been observed in the spinal cord and dorsal root ganglion neurons innervating regions where tumor and peripheral nerves interact. In a murine model of bone cancer using sarcoma cells, MOR expression was downregulated in the spinal dorsal horn and dorsal root ganglion (DRG) neurons, changes corresponding to decreased morphine sensitivity [58]. Decreased MOR expression was identified in DRG neurons that co-expressed MOR and TRPV1 or CGRP, indicating that decreased MOR influences nociceptive processing in murine bone cancer pain [59]. Reduced morphine sensitivity necessitating higher doses further complicates treatment, as morphine itself inhibits NK cells and activates mast cells, simultaneously blocking anticancer immune

activity and generating cancer-promoting and pro-inflammatory cytokines that in turn increase nociceptor activity, such as prostaglandins and substance P [44]. Inhibition of COX-2 during morphine administration in a murine breast cancer model not only blocked COX-2 and $PGE_2$ release but also decreased angiogenesis and tumor progression while prolonging survival and analgesic efficacy [60]. Conversely, MOR expression was increased in DRG neurons of rats implanted with breast carcinoma bone cancer [61], indicating that there may be cancer- and species-specific changes in MOR expression associated with bone cancer development.

### 2.2. MOR Opioids and the Immune System

MOR is present in immunocytes, such as natural killer cells or other lymphocytes, neutrophils, and macrophages [62–65]. The effect of MOR agonists on immune cell functions varies according to their structural class, dose, and duration of exposure [66–68]. For instance, short-term treatment with fentanyl and methadone can stimulate the production of interleukin (IL-4) 4 in T cells, which is suppressed in the presence of morphine [67]. Morphine also suppresses the production of IL-2 in those cells and stimulates the function of T regs while suppressing that of Th17 [69–71]. Although MOR is responsible for the immunosuppressive effect of morphine and has been linked to TCR-activated signaling and decreased activities of AP-1 and NF-Kβ, a neural mechanism has also been implicated in morphine-mediated immunosuppression [71,72].

Immune checkpoint inhibitors (ICIs) are reshaping the treatment strategies of cancer patients [73]. Recent evidence suggests that opioids could negatively impact the effect of immunotherapies [74–76]. Kostine et al. showed that patients who were treated with morphine had a decreased rate of tumor response to ICIs compared to controls (51% versus 71%) [76]. A systematic review by Cani et al. analyzed 13 retrospective observational studies [77]. Their review demonstrated a negative correlation between ICIs and opioid use with respect to progression-free survival and overall survival. While the results of Cani's review are concerning, since these studies were retrospective and heterogeneous, the authors concluded that there is still a lack of solid evidence to avoid the prescription of opioids in patients receiving ICIs [77].

## 3. Prescription Opioids in Patients with Cancer

In 1914, the United States government recognized the potential for opioid abuse and misuse and approved the Harrison Narcotics Act, which prohibited the use of prescription opioids for nonmedical use [78]. A century later, the so-called "opioid epidemic" was declared in the United States and has spread among high-income countries. The prescription opioid epidemic represents a significant public health crisis, highlighted by the drastic increase in opioid prescriptions and misuse [79]. In the United States, 191 million prescriptions were dispensed in 2017, with misuse reported by approximately 10.1 million people in 2019 (https://www.cdc.gov/nchs/nvss/vsrr/drug-overdose-data.htm, accessed on 13 March 2024). This over-prescription has led to widespread opioid availability, contributing to over 300,000 opioid-related hospitalizations and nearly 50,000 overdose deaths annually (https://www.ahrq.gov/opioids/map/index.html, accessed on 13 March 2024). The economic impacts are profound, with an estimated cost in the U.S. alone of over USD 78.5 billion annually in healthcare, lost productivity, and associated societal costs [80]. The epidemic's scale underscores the urgent need to reevaluate prescribing practices. This reevaluation aims to balance necessary pain management for conditions like cancer against the risks of misuse and addiction. Strategies to mitigate this crisis include enhancing prescribing guidelines, boosting public awareness, and improving access to addiction treatment services [81].

The prevalence of pain among cancer patients is high; in a systematic review, the reported rates of pain reached 55% during anticancer treatment and 39% after curative treatment [2]. Moderate-to-severe pain, which can be considered an indication for opioid prescription, was reported by 38% of all patients. Expectedly, a large study conducted on

the veteran population found that cancer patients were almost twice as likely to have an opioid prescription compared with noncancer patients [82]. These numbers raise a concern regarding the risk of developing opioid use disorders such as dependency or misuse among cancer patients and survivors [83,84].

### 3.1. Epidemiological Studies on Prescription Opioids and Cancer Formation

Despite all the proposed mechanisms and the translational research described above, the findings of clinical and epidemiological studies regarding opioids and cancer formation are somewhat inconclusive and mostly focus on opioid drug abusers (e.g., opium, heroin). For instance, an ecological study using global registries of cancer, drugs, and crime found that an increased prevalence of opiate use in a country was significantly associated with an increased incidence of bladder, kidney, oral cavity, esophagus, laryngeal, and pharyngeal cancers [85]. In a systematic review by Mansouri et al., 21 observational studies comparing ever-opium users with never-opium users were reviewed [86]. Altogether, 64,412 subjects with 6658 cases of cancer were analyzed, and opium users were found to have a 3.53 times greater risk for overall cancer. This positive association was also observed by cancer type (e.g., gastrointestinal, bladder, head and neck, larynx, and lung), with higher opium doses and prolonged duration of consumption correlating with further increased risk. A large cohort followed 18,659 individuals chronically prescribed methadone for opioid drug dependency (e.g., heroin) for ten years. The study found a substantially increased risk for developing cancers such as lung, larynx, and liver, while having a lower risk for colorectal and breast cancer [87].

While the correlation between opioid drug abuse and cancer formation was investigated and supported by multiple epidemiological studies, drawing direct conclusions from these studies is difficult. Opium or opioid recreational drug users constitute a unique population group with distinct common characteristics such as heavy tobacco use, medical negligence, as well as specific comorbidities and socioeconomic factors. Therefore, this group displays confounding factors that are challenging to completely adjust for and may potentially contribute to the increased risk of cancer observed in this group. In this context, epidemiological studies on prescription opioid users, a different population group who do not share the same characteristics, are of great value. The epidemiological research on prescription opioid drug users and the incidence and development of cancer is sparse, yet all available data can significantly contribute to our understanding of the relationship between opioids and the risk of cancer. These studies generally utilize large datasets of noncancer patients treated with prescription opioid drugs for pain to derive correlations, incidences, and hazard ratios.

Boudreau and colleagues investigated a specific group of patients with an increased risk of developing cancer, patients who were previously diagnosed with primary breast cancer [88]. Data were obtained from the medical record review, the SEER tumor registry, and electronic health records. Their cohort included 4216 patients previously diagnosed with breast cancer, of whom 143 experienced second primary breast cancers during follow-up, and 410 were defined as chronic prescription opioid users. This study reported a multivariable-adjusted hazard ratio for chronic opioid users to develop a second primary cancer of 1.38, compared to non-opioid users. However, this reported risk was insignificant (95% CI: 0.71–2.70). Hence, the authors of this study concluded that the findings of their study provide some reassurance on safety regarding chronic opioid use and cancer, yet they recommended further exploration in other populations and settings. Indeed, few epidemiological studies published in the next few years were able to provide more indicative findings on this matter.

Another study was a population-based cohort by Oh and Song, who used data from the South Korean National Health Insurance Service [89]. This study included 351,701 patients diagnosed with "musculoskeletal system and connective tissue diseases", of whom 25,153 were prescribed opioids for $\geq$90 days. On a multivariate model, the adjusted hazard ratio (aHR) for developing cancer among chronic opioid users was 1.20 (95% CI: 1.15–1.25). The

authors also conducted a subgroup analysis according to opioid potency and found higher aHR for strong opioids, such as fentanyl, morphine, oxycodone, hydromorphone, and methadone. The aHR results for weak and strong opioid users were 1.18 (95% CI: 1.13–1.23) and 1.32 (95% CI: 1.10–1.59), respectively.

A more recent study by Sun et al. obtained noncancer patients' data from the Taiwan National Health Insurance Research Database between 2008 and 2019. Patients with chronic pain who were never previously diagnosed with cancer and were prescribed opioids for pain management (at least 180 doses annually and over three months) comprised the case group, which included 50,888 patients [90]. A 4:1 ratio control group (12,722 subjects) was matched for various potential confounders, including age, sex, sleep disorder, alcohol consumption, smoking, income level, urbanization, and a list of various comorbidities. The results of a multivariate analysis indicated that patients in the opioid group had a higher risk of developing cancer with an aHR of 2.66 (95% CI: 1.44–2.94) compared with the non-opioid group. Data analysis specified according to cancer types also demonstrated increased aHR for several types of cancer, including lung (2.63), hepatocellular (2.63), colorectal (3.13), breast (3.23), prostate (2.85), head and neck (2.22), pancreatic (1.52), gastric (3.23), esophageal (2.5), and ovarian (3.03) cancers.

Another case–control study by Havidich et al. utilized the SEER tumor registry and linked it with Medicare's prescriptions data [91]. The study population (N = 348,319) included 143,921 cancer patients who were matched with 204,398 controls, of whom, 34% were prescribed with opioids. The study found conflicting results: patients who were exposed to prescription opioids had a lower adjusted odds ratio (aOR) for breast and colon cancer, while having a high aOR for leukemia, lymphoma, renal, lung, and liver cancer.

A recent pre-published study (Sheikh M, Alcala K, Mariosa D, et al. Regular use of pharmaceutical opioids and subsequent risk of lung cancer. WCLC 2023, 9–12 September 2023), presented at the 2023 World Conference on Lung Cancer, analyzed the 473,067 participants of the UK Biobank study who were recruited between 2006 and 2010 and were cancer-free on recruitment. Participants were followed up to April 2021, and 3480 of them developed lung cancer during follow-up. At baseline, 27,856 participants reported on regular opioid prescription drug use. Following adjustment for potential confounders, regular opioid use was found to be associated with an increased aHR for developing lung cancer of 1.32 (95% CI: 1.20–1.47). Their results also indicated that the risk for lung cancer increased in a dose-dependent manner with higher potency and longer duration of action of opioids used, reaching 1.82 (95% CI: 1.29–2.57) and 1.85 (95% CI: 1.29–2.64) aHR among participants using potent and long-acting opioids compared to those not regularly using opioids. These risks were comparable across strata of sex, chronic pain, smoking, alcohol use, and socioeconomic status.

Finally, another pre-published work is a case–control study on 22,616 subjects using data from the Clalit Health Maintenance Organization District Registry in Israel. In this study, 11,308 cancer patients were matched with an equal number of cancer-free controls. Matching was based on multiple factors such as age, gender, socioeconomic status, BMI, smoking habits, alcohol abuse, and specific medical conditions. Prescription opioid usage was reviewed during the decade before the cancer diagnosis. A multivariate logistic regression analysis revealed that long durations of opioid use were associated with an increased risk of cancer, with an annual aHR of 1.03 (95% CI: 1.01–1.05), suggesting that each additional year of opioid use increases cancer risk by 3%.

### 3.2. Opioids in the Context of Cancer Surgery

Despite an increasing trend in the use of multimodal opioid sparing analgesia strategies in patients with cancer undergoing surgery, opioids remain the standard agents for rescue from moderate-to-severe pain. It has been speculated that high dosages of opioids or their persistent use after oncological surgery could influence the prognosis of patients [92]. For instance, Nelson et al. showed in a cohort of patients with stage 1 non-small-lung

cancer a strong association between prescription opioid use 3 to 6 months after surgery and worse survival [93].

Based on the premise that opioids could negatively influence prognosis in patients with cancer, opioid-free anesthesia (OFA) and opioid-sparing anesthesia techniques have been studied, described, and implemented [94,95]. Only one randomized controlled trial studied the effect of OFA on cancer progression after surgery. The study showed that the avoidance of opioids during radical prostatectomy did not improve biochemical-free survival. [96] In different clinical trials, cancer recurrence was not impacted by a moderate reduction in opioid consumption during cancer surgery [97,98].

## 4. Conclusions

The current preclinical and clinical evidence described above highlights the complex interplay between opioid use and cancer. The preclinical data support the role of MOR in modifying cell behaviors. Epidemiological data suggest a possible association between prescription opioid usage and cancer development (Table 1) [4,99–109]. Nevertheless, the role and significance of prescription opioid usage in cancer formation is still not completely understood, and careful consideration in future studies should be given to the specific type of cancer and immune status, specific drug used, dosage, and duration of usage. Several incidences indicate the involvement of opioids in cancer, yet we need more dose-dependent, controlled experimental data to clearly conclude the involvement of the opioids in the cancer progression at the prescribed dose for pain management. While prescribing opioids for pain management, the side effects should be considered.

**Table 1.** Dose dependent effect of opioids on cancer.

| Author/Year/Ref # | Type of Cancer | Opioids | Dose | Findings |
|---|---|---|---|---|
| Forget et al, 2011 [99] | Prostate cancer | Sufentanil | Mean sufentanil dose: 23 μg | Increased risk of recurrence (HR: 7.78; 95% CI: 5.79–1.78) |
| Cata et al., 2014 [100] | Non–small cell Lung cancer | Morphine | Median morphine equivalents: 1358.6 mg | No impact on RFS (HR: 1.074 CI 95%:0.989–1.166) No impact on OS (HR: 1.06 CI 95%:0.964–1.165) |
| Oh et al., 2017 [101] | Non–small cell Lung cancer | Morphine | Median morphine equivalents: 819 mg | Increased risk of recurrence (aHR: 1.415; 95% CI: 1.123–1.781) Higher mortality (aHR: 1.514; 95% CI: 1.197–1.916) |
| Oh et al., 2017 [102] | Esophageal SCC | Remifentanil, Morphine, hydromorphone, fentanyl, oxycodone, | Morphine equivalent 10 mg | Increase risk of reoccurance. (aHR, 1.274; 95% CI: 0.922–1.761;) |
| Maher et al., 2019 [103] | Non–small cell Lung cancer | Morphine | Mean morphine equivalents: 124 vs. 232 mg | Increased risk of recurrence (OR: 1.003; 95% CI: 1.000–1.006) |
| Cata et al., 2014 [100] | Laryngeal SCC | Fentanyl | Median fentanyl equivalents: 526 μg | Shorter RFS (HR: 1.001; 95% CI: 1.00–1.001) Shorter OS (HR: 1.001; 95% CI: 1.00–1.001). |
| Patino et al., 2017 [104] | Oral cancer | Fentanyl | Median fentanyl equivalents: 1081 μg | No impact on RFS (HR: 1.27; CI 95%:0.838–1.924) Shorter OS (HR: 1.77; CI 95%: 0.995–3.149] |
| Owusu-Aygemang et al., 2018 [105] | Pediatric Abdominal Malignancies | Morphine | Median morphine equivalents: 18.9 mg | No impact on RFS (HR: 1.00; 95% CI: 0.99–1.02) No impact on OS (HR; 1.01; 95% CI: 0.99–1.03) |

**Table 1.** *Cont.*

| Author/Year/Ref # | Type of Cancer | Opioids | Dose | Findings |
|---|---|---|---|---|
| Tai et al.,2017 [106] | Colorectal cancer | Fentanyl | Mean fentanyl dose: 3 μg/kg | No impact on RFS (aHR: 0.93; 95% CI: 0.74–1.17) No impact on OS (aHR: 0.79; 95% CI: 0.52–1.19) |
| Cata et al., 2015 [107] | Non-small cell lung cancer | Fentanyl, sufentanil, remifentanil | Fentanyl equivalents >28.2 μg/kg | Impact survival (aHR: 0.779, 95% CI: 0.619–0.980 ) |
| Patino et al., 2017 [104] | Oral cancer | Fentanyl, sufentanil, remifentanil, morphine, hydromorphone | Fentanyl equivalent 1 μg/kg | Higher mortality risk (aHR: 1.27, 95% CI: 0.838–1.924) No impact on OS (aHR: 1.77, 95% CI: 0.995–3.149) |
| Du et al., 2018 [108] | Esophageal cancer | Fentanyl, sufentanil Remifentanil and hydromorphone | Fentanyl equivalent 1 μg of fentanyl were as follows: 0.1 μg of sufentanil, 1 μg of remifentanil, and 10 μg of hydromorphone | Better RFS (aHR: 0.376, CI 95%: 0.201–0.704) Improved OS (aHR: 0.346, CI 95%: 0.177–0.676) |
| Sathornviriyanpong et al.,2016 [109] | Advance Cancer | Morphine | Morphine equivalent 6.43 mg/day | No impact ≤30 vs >30 mg/day (aHR: 1.14, 95% CI: 0.77–1.69) |
| Zylla et al.,2014 [4] | Prostate | Oxycodone, hydrocodone, codeine, morphine, hydromorphone fentanyl and methadone | Morphine equivalent 5mg/day | Impact survival (aHR: 0.92, 95% CI: 0.68–1.25) |

Adapted from Ramirez et al. [58]. aHR indicates adjusted hazard ratio; CI, confidence interval; HR, hazard ratio; OS, overall survival; RFS, recurrence-free survival; SCC, squamous cell carcinoma.

**Author Contributions:** Conceptualization, H.S.-H., J.P.C. and V.G.; writing—original draft preparation, D.S., N.T., M.L.U., H.S.-H., J.P.C. and V.G.; writing—review and editing, D.S., M.L.U., J.P.C., H.S.-H., N.T. and V.G.; visualization, D.S., M.L.U. and J.P.C.; supervision, J.P.C., V.G. and H.S.-H.; project administration, J.P.C. All authors have read and agreed to the published version of the manuscript.

**Funding:** This research received no external funding.

**Conflicts of Interest:** The authors declare no conflict of interest.

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
