# Peer review of "Opioids and Cancer: Current Understanding and Clinical Considerations"

_curroncol, doi:10.3390/curroncol31060235_

Round 1

Reviewer 1 Report

Comments and Suggestions for Authors

Dear Authors,

I have read the manuscript and I even if that the data are of clinical interest I have not well understand your conclusions.

In your manuscript you describe the association between drug use and cancer development. In particular you report that 1 year of opioid use induces a 3% increased risk of cancer. In conclusion you write "prescription opioid usage in cancer formation is still not completely understood,"...."patients with cancer with moderate-to-severe pain may receive opioids to decrease suffering".

 I think that this is not scientific and not reasonable particularly in a clinical setting.

The conclusions may report the data of the manuscript if you report an increased risk you can not write that a cancer patient can use opioids.

Moreover tables must be added reporting the data in humans and in animals.

Finally please describe in a table the dosage of opioid induced an increase in cancer development.  

Comments on the Quality of English Language

none

Author Response

Reviewer #1

I have read the manuscript and I even if that the data are of clinical interest I have not well understand your conclusions.

Response: I am thankful to reviewer for this suggestion. In this manuscript we explain about the side effects of the prescription opioids on cancer progression, conclusion is modified and table added.

Change in the manuscript: Conclusions modified; table added.

In your manuscript you describe the association between drug use and cancer development. In particular you report that 1 year of opioid use induces a 3% increased risk of cancer. In conclusion you write "prescription opioid usage in cancer formation is still not completely understood,"...."patients with cancer with moderate-to-severe pain may receive opioids to decrease suffering".

Response: I am thankful to reviewer for this suggestion. Cited evidence support the association of drug with cancer but make firm conclusion we need more data from dose dependent controlled experiments.

Change in the manuscript: Conclusion section modified

I think that this is not scientific and not reasonable particularly in a clinical setting.

Response: It can be use as guidance while prescribing opioids, while ongoing research focused to generate more dose dependent experimental data.

Change in the manuscript: Conclusion section modified

The conclusions may report the data of the manuscript if you report an increased risk you cannot write that a cancer patient can use opioids.

Response: Opioids are one of the standard cares of pain for cancer patients with moderate to severe pain.

Change in the manuscript: NA

Moreover tables must be added reporting the data in humans and in animals.

Response: I am thankful to reviewer for this suggestion, a table is added.

Change in the manuscript: Table added

Finally please describe in a table the dosage of opioid induced an increase in cancer development.  

Response: I am thankful to reviewer for this suggestion, a table is added.

Change in the manuscript: Table added

Reviewer 2 Report

Comments and Suggestions for Authors

The manuscript by Sah et al. reviews literature on opioids in cancer with focus on the role of the mu-opioid receptor in cancer progression. Although the topic of the review manuscript is of relevance, there were some issues primarily to the introduction that need to be addressed by the authors.

Introduction

On page 2, the authors write “Opioids vary in their origin and potency”. What do they mean by origin? Natural, synthetic?

The authors are requested to elaborate on the differences in the potencies and efficacies of most relevant clinically used opioids in treatment of cancer pain, with respective references.

It would be valuable for the readers to show the chemical structures of the opioids discussed.

There is only description on the therapeutic analgesic effect of opioids, but no reference is made to the adverse effects of opioids following acute and chronic treatment. This should be revised. A useful, recent reference to the side effects of opioids is doi: 10.3390/ph14111091.

Title of some sections should be revised:

Section 2 – MOR biology in cancer cells.

Section 2.1. – MOR and nerve-cancer cell interaction; also in the first line of the paragraph.

Comments on the Quality of English Language

Minor corrections requires, Some type errors. 

Author Response

The manuscript by Sah et al. reviews literature on opioids in cancer with focus on the role of the mu-opioid receptor in cancer progression. Although the topic of the review manuscript is of relevance, there were some issues primarily to the introduction that need to be addressed by the authors.

Introduction

On page 2, the authors write “Opioids vary in their origin and potency”. What do they mean by origin? Natural, synthetic?

Response: I am thankful to reviewer for this suggestion. We mean the as you suggested natural, synthetic, semi-synthetics.

Change in the manuscript: Explanation added to section.

The authors are requested to elaborate on the differences in the potencies and efficacies of most relevant clinically used opioids in treatment of cancer pain, with respective references.

Response: I am thankful to reviewer for this suggestion, a table added with different dose and response.

Change in the manuscript: Table added

It would be valuable for the readers to show the chemical structures of the opioids discussed.

Response: I am thankful to reviewer for this suggestion, we cited the manuscripts which discussed the structure and chemistry of opioids. We primary focused on the overall effect of opioids on cancer irrespective of their structure.

Change in the manuscript: No change

There is only description on the therapeutic analgesic effect of opioids, but no reference is made to the adverse effects of opioids following acute and chronic treatment. This should be revised. A useful, recent reference to the side effects of opioids is doi: 10.3390/ph14111091.

Response: I am thankful to reviewer for this suggestion, we added the reference.

Change in the manuscript: Reference added to the manuscript

Round 2

Reviewer 2 Report

Comments and Suggestions for Authors

The authors have addressed my concerns.